materials science/nanotechnology/environmental science

engineered nanomaterials, occupational health, quantum dots, contact transfer, simulated spills

**Author for correspondence:**
Byron D. Gates
e-mail: bgates@sfu.ca

†These authors contributed equally to this study.

This article has been edited by the Royal Society of Chemistry, including the commissioning, peer review process and editorial aspects up to the point of acceptance.

# Contact transfer of engineered nanomaterials in the workplace

Irene Andreu[†], Tuan M. Ngo[†], Viridiana Perez, Matthew W. Bilton, Kelly E. C. Cadieux, Michael T. Y. Paul, Tania C. Hidalgo Castillo, Clifton Bright Davies and Byron D. Gates

Department of Chemistry and 4D LABS, Simon Fraser University, 8888 University Drive, Burnaby, British Columbia, Canada V5A 1S6

IA, 0000-0001-7689-2269; VP, 0000-0002-8660-2656; BDG, 0000-0001-9108-3208

This study investigates the potential spread of cadmium selenide quantum dots in laboratory environments through contact of gloves with simulated dry spills on laboratory countertops. Secondary transfer of quantum dots from the contaminated gloves to other substrates was initiated by contact of the gloves with different materials found in the laboratory. Transfer of quantum dots to these substrates was qualitatively evaluated by inspection under ultraviolet illumination. This secondary contact resulted in the delivery of quantum dots to all the evaluated substrates. The amount of quantum dots transferred was quantified by elemental analysis. The residue containing quantum dots picked up by the glove was transferred to at least seven additional sections of the pristine substrate through a series of sequential contacts. These results demonstrate the potential for contact transfer as a pathway for spreading nanomaterials throughout the workplace, and that 7-day-old dried spills are susceptible to the propagation of nanomaterials by contact transfer. As research and commercialization of engineered nanomaterials increase worldwide, it is necessary to establish safe practices to protect workers from the potential for chronic exposure to potentially hazardous materials. Similar experimental procedures to those described herein can be adopted by industries or regulatory agencies to guide the development of their nanomaterial safety programmes.

## 1. Introduction

Research laboratories around the world are investigating engineered nanomaterials for a variety of applications, spanning

from enabling clean energy technologies to improving our ability to effectively treat cancer [1,2]. Due to the special properties of engineered nanomaterials, more than 1000 nanomaterial-containing consumer products are currently available on the market [3]. It is expected that the global market for engineered nanomaterials will continue to increase in the near future [4]. One estimation suggested that there could be as many as 6 million workers worldwide who handled engineered nanomaterials in 2020 [1]. Nanomaterials can differ in their size, shape, composition and surface coating, and each of these factors can influence the toxicity profile of a specific engineered nanomaterial [5,6]. Although much progress has been made in assessing the health effects of specific engineered nanomaterials, the toxicity of nanomaterials should be evaluated on a case-by-case basis due to the wide variety of compositions and properties [5,6]. While a large body of literature focuses on the toxicological effects of nanomaterials in human cell cultures and their ecological implications, few reports in the literature studied the exposure and health hazards for workers during fabrication and handling of engineered nanomaterials [7–9]. Due to the lack of safety information, a widespread recommendation when handling engineered nanomaterials is to minimize exposure through the use of appropriate engineering controls and personal protective equipment [10–12].

To determine if the appropriate engineering controls and personal protective equipment have been put in place, it is important to understand the propagation of engineered nanomaterials in the workplace during routine handling or after a spill. The generation of aerosols containing engineered nanomaterials has been found to be an important propagation pathway in the workplace, and inhalation is the most studied exposure route [8,11]. Recently, Clemente et al. used fluorescent engineered nanomaterials to demonstrate how these materials can propagate through the air during common workplace practices like pouring powders [13]. Contact transfer is another potential pathway for engineered nanomaterial propagation in the workplace. Contact transfer can occur when a surface containing engineered nanomaterials, e.g. a benchtop after a spill or a dirty pipette, comes into physical contact with a clean surface, e.g. a glove or a hand. The engineered nanomaterials can then be transferred to additional clean surfaces by subsequently contacting said glove or hand with these other surfaces (i.e. resulting in secondary contact transfer). Additional secondary contact transfers can potentially occur, spreading these engineered nanomaterials throughout the workplace. Previous research has shown that powders and liquids can transfer to bare hands when handling them and can transfer from hands to the mouth via contact transfer, indicating a potential dermal or even ingestion exposure [14,15]. In spite of this evidence, contact transfer as a contamination spread pathway has not been studied in detail for engineered nanomaterials [14,16,17]. Few studies have dealt with the transfer of engineered nanomaterials via surface-to-surface contact in a workplace setting. For example, this study is relevant to a situation where the worker might be applying pressure with their gloved hand onto a surface that contains engineered nanomaterials. Brouwer et al. showed that large amounts of ZnO powder were transferred to bare hands when the powder contained nanosized particles. The transfer for powders containing micron-sized particles of the same sample composition was significantly lower [14]. This study also found that the transfer efficiency is higher for powders deposited on a metallic surface than on wood, hinting at the potential influence of the substrate roughness and hydrophobicity. The contact transfer literature has focused, so far, on a direct transfer to the exposed skin or to the area around the mouth [15]. Workers handling nanomaterials are expected to wear gloves in workplace settings as recommended by several occupational safety guidelines [10,12]. The transfer of engineered nanomaterials by means of contact with gloves is, therefore, of particular interest. To the best of our knowledge, the work of Gorman Ng et al. [15] is the only published article to report a powder being transferred from contaminated surfaces to gloves via contact transfer. Previous research has shown that engineered nanomaterials can accumulate in the pores of gloves [11,18], but these studies were performed by incubating glove pieces with nanosized materials without applied pressure. The study of contact transfer of engineered nanomaterials under conditions representative of realistic situations, and between relevant surfaces (e.g. gloves and countertops), is needed to improve risk assessment in the workplace. An experimental set-up to reproducibly study the processes involved in contact transfer is also necessary to obtain reliable data. The composition, surface chemistry, size and shape of the nanomaterials are anticipated to influence the contact transfer process [14]. However, both Brouwer et al. and Ahn et al. did not describe the surface chemistries of the engineered nanomaterials used in their studies [14,18].

We have focused on the contact transfer of engineered nanomaterials between contaminated countertops (CTs) to clean gloves (primary transfer) and the contact transfer between said gloves and other pristine substrates (secondary transfer) as shown in figure 1a. To control for the effects of nanomaterial physicochemical properties on the contact transfer process, we used one type of nanomaterial for the entire study. We chose nitrile gloves as vehicles between the primary and

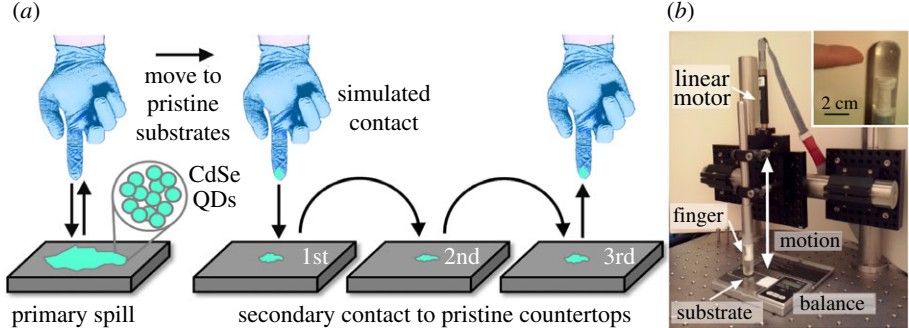

**Figure 1.** (*a*) Schematic depiction of the experimental procedure used for assessing the contact transfer of engineered nanomaterials (i.e. cadmium selenide (CdSe) quantum dots (QDs)) to a gloved finger and subsequently to a series of pristine substrates and (*b*) the experimental set-up used to study the probability for contact transfer. Inset: picture of the tip of the simulated finger and, for comparison, an adult human index finger.

secondary transfer due to their ubiquitousness in the laboratory and high chances of contacting other surfaces compared with other potential recipients of the primary transfer, such as the bottom of glassware. Cadmium selenide quantum dots (CdSe QDs) with a hydrophobic surface coating were chosen as our model engineered nanomaterial. We chose CdSe QDs because QDs are commonly used in several industries and research laboratories due to their photoluminescence and photoelectric properties [19]. QDs are often composed of toxic elements such as cadmium or lead [20]. Cadmium is known to affect renal function and result in skeletal damage [21,22]. It is also a suspected carcinogen [23]. The Food and Agriculture Organization of the United Nations and the World Health Organization have estimated that the tolerable weekly intake of cadmium for an adult is between 400 and 500 µg [24]. The toxicity of CdSe in the form of nanomaterials is still being quantified, but it is anticipated to be equal to or higher than that of micron-size powders of the same composition [25,26] due to both the smaller dimensions and the high ratio of surface area to mass for nanomaterials relative to micron-size or larger materials. The contact transfer process was carried out using an electronically controlled mechanical system to reduce experimental error and human variability, in contrast to previously published contact transfer studies [14] where human subjects used their hands or other body parts to initiate contact with contaminated surfaces. In this study, the contact transfer of CdSe QDs to different surfaces, which are representative of situations in which nanomaterials would be handled, is evaluated both qualitatively by optical inspection and quantitatively for selected surfaces using inductively coupled plasma mass spectroscopy (ICP-MS) techniques. Understanding the potential for propagation of engineered nanomaterials between surfaces in workplace settings and their detection and quantification is critical to inform occupational health and safety decisions, and to foster a safe environment for workers.

# 2. Methods

## 2.1. Synthesis and characterization of quantum dots

The oleic acid-coated CdSe QDs used for this study were synthesized following a previously reported procedure [27]. The experimental details for the synthesis and characterization of these QDs can be found in the electronic supplementary material.

## 2.2. Simulation of spills and contact transfer events

Special care was used in the experimental design to achieve a reproducible contact with different substrates. A mechanical test rig was established and is shown in figure 1*b*. A simulated finger was made of a soft, flexible polymer (PDMS Dow Corning Sylgard® 184, USA), moulded with dimensions and curvature of the tip that reproduce those of adult human fingertips (inset, figure 1*b*). The simulated finger was covered with a pristine nitrile glove before contact with each of the simulated primary spills (Aurelia® Robust® nitrile gloves, 4.5 mil, medium size, USA). The simulated gloved finger was positioned in a vertical direction for a controlled approach and retraction from each of the substrates under test. A DC linear servo motor actuator was used (Thorlabs Inc. Z625B 25 mm motorized actuator with DX-PCI100 controller) to actuate the simulated finger (figure 1*b*). Substrates

were placed one at a time on a digital balance located below the finger, and the motor was controlled by software (MCAPI from Precision MicroControl Corporation) for a high degree of accuracy and reproducibility. The balance was used to measure the pressure load on each substrate during contact with the gloved finger. Once a desired load was achieved, the finger was retracted in the vertical direction to avoid any shearing forces. Contact with each substrate at the desired load was approximately 1 s. The contact transfer experiments were performed at ambient conditions in a laboratory with minimal airflow currents and temperature or humidity fluctuations, as these could affect the contact transfer results. The described set-up and experimental procedure enabled us to simulate hand contact with the original spill and secondary substrates with a high degree of reproducibility of the applied force, approach angle and duration of the contact.

The primary spills were prepared by drop casting different amounts of CdSe QDs from a toluene suspension onto approximately 1 cm$^2$ sections of a representative CT found in several laboratories at Simon Fraser University (Wilsonart high-pressure laminate Chemsurf Black 1595 Matte 60). The contaminated CTs containing CdSe QDs were left to dry at ambient conditions in clean, closed plastic Petri dish boxes, for periods of time between 30 min and 7 days after applying the spill to analyse the influence of drying time on the contact transfer. The protocol for evaluating QD contact transfer between the finger and the substrate begins with the transfer of the QDs from the primary spill to the nitrile glove (primary transfer). The contaminated gloved finger was subsequently pressed against clean substrates (secondary transfer) to assess the potential for continued, sequential transfer from the glove. Each test for secondary transfer was prepared by contacting a new clean substrate with the contaminated gloved finger to create many successive 'prints' originated from a single primary transfer event. The substrates tested for the secondary transfer of QDs were representative of those surfaces commonly found in laboratories that handle engineered nanomaterials. These substrates were CTs, nitrile gloves, disposable laboratory coats, keys on a computer keyboard, curved stainless steel rods (representative of door handles) and writing paper. The CTs, gloves, laboratory coats and paper were cut into approximately 1 cm$^2$ sections. All substrates were immobilized on glass slides using double-sided tape before initiating contact transfer with the substrates as outlined earlier, for ease of handling. All suspensions of the QDs were manipulated in a fume hood. Substrates containing QDs were kept away from light to minimize their potential influence on the properties or to induce photodegradation of the QDs.

## 2.3. Qualitative investigation of the simulated spills and transferred materials

Before and after contact with transferred material from the initial spills of QDs, the substrates were assessed for the presence of light-emitting QDs by qualitative methods. These substrates were imaged under ultraviolet (UV) light illumination (peak excitation wavelength 254 nm, output power 8 W) to monitor for the presence of these engineered nanomaterials. Pictures were acquired using a 12-megapixel camera equipped with a f/1.7 lens on a Samsung S7 Edge smartphone maintained at a distance of approximately 10 cm from each substrate and at a fixed angle of approximately 45° between the substrate and the camera.

## 2.4. Quantitative analyses of the simulated spills and residue from contact transfer processes

The amounts of Cd present on the sections of the CT after the described procedure for initiating primary and secondary contact transfer were determined by ICP-MS. These results were compared with those for pristine CTs and primary CT spills. The same nominal volume from the suspension of QDs used to create the simulated spills was also added directly into a series of glass vials to assess variance in the quantity of QDs samples from the original suspension. The samples were each handled carefully to avoid cross-contamination. All samples were digested in clean glass vials using 2 ml of concentrated nitric acid (nitric acid, TraceSELECT® Ultra less than 0.1 ppb metal content) at 60°C for 2 h. After this acid digestion process, aliquots of the resulting acidic solution were diluted with 18 MΩ cm deionized water to achieve a 500x dilution factor with a final concentration of 2% (v/v) HNO$_3$. These diluted acidic solutions were filtered through 0.2 μm polytetrafluoroethylene (PTFE) filters to remove any residual solids from the digestion process, such as fibres from the CT. The filtered, diluted solutions were analysed by ICP-MS using a ThermoFisher iCAP Qc ICP-MS instrument equipped with a DX FAST autosampler and a peristaltic pump, operated by iCAP Q and Qtegra software. Argon gas was used as the carrier gas for these measurements, and all of the samples were analysed using the kinetic energy discrimination mode with an internal standard of [115]In obtained from a commercially available

multi-element solution (i.e. Tune B iCAP Q from Thermo Fisher) containing $1.0\,\mu g\,l^{-1}$ of a series of elements (i.e. Ba, Bi, Ce, Co, In, Li and U) in a solution of 2% (v/v) $HNO_3$ and 0.5% (v/v) HCl. The method for operating the ICP-MS used a dwell time of 0.03 s and an integration for 20 sweeps at a normal resolution (the quadrupole was operated between 0.7 to 1 amu) per run. This method was repeated for five runs per sample, and the results were integrated for individual samples. The argon gas flow rate of the nebulizer was optimized between 0.38 and $0.46\,ml\,min^{-1}$. The ICP-MS instrumentation was also located inside a clean hood (ultra-low penetration air or ULPA filtered, continuous airflow) to minimize further risks of sample contamination during these analyses. For each set of digestions and set of samples analysed by ICP-MS, a series of procedural control samples were also prepared and processed in parallel to account for variations in the glassware and solvents and to determine the background levels for each analysis. The amount of cadmium present in the solutions was quantified using standard calibration curves with cadmium concentrations ranging from 0.1 to 10 ppb, which were prepared from ICP-MS standards (Sigma-Aldrich). Cadmium was selected over selenium as the element to monitor for these quantitative measurements due to potential interferences between the signal from the selenium species and the argon carrier gas used in the ICP-MS analyses [28]. Three series of primary and secondary transfers were analysed by ICP-MS. For the pristine CT control, six samples were analysed by ICP-MS to accurately quantify the baseline for the amount of Cd in the samples. The total amount of cadmium present in each of the samples was calculated after accounting for the dilution factor and the total volume of the digestion solutions. Mean values were calculated from the replicate measurements, and errors were reported as one standard deviation from these mean values.

# 3. Results and discussion

## 3.1. Characterization of the nanomaterials

The evaluation of the potential for contact transfer of engineered nanomaterials used CdSe QDs as a model system. The non-polar nature of the coating on these QDs imparts a hydrophobic property to these nanomaterials. This property improves their stability in non-polar organic solvents such as hexanes and toluene. These hydrophobic QDs are representative of many engineered nanomaterials under study around the world. For example, oleic acid, coating for the present QDs, is a common surfactant used to stabilize many engineered nanomaterials such as magnetic nanoparticles [29–31]. The findings of this evaluation for contact transfer could be relevant for other nanomaterials with similar surface coatings. The size and shape of the QDs used for this study were characterized by transmission electron microscopy (TEM; inset, figure 2a). These nanoparticles were approximately spherical, with diameters between 3 and 5 nm. The image in figure 2a shows the fluorescence emission spectrum for a suspension of QDs in toluene with an incident excitation of 400 nm. QDs have a maximum fluorescence emission at 540 nm (a green fluorescence). The presence of QDs within simulated spills and residue transferred to various substrates was, therefore, easily confirmed through a visual inspection of the substrates under UV illumination (figure 2c,e). The green fluorescent emission of the QDs revealed their presence on the substrates. These same simulated spills and transferred residue containing QDs on the various substrates would otherwise be difficult to discern under regular, white light illumination (figure 2b,d). Not all nanomaterials are intrinsically fluorescent, and it would, therefore, not always be possible to evaluate the presence of nanomaterials transferred by contact with surfaces in the laboratory through the use of UV-induced fluorescence emission. The visible fluorescence emission of the QDs assisted in tracking these nanomaterials at each stage of the processes used to evaluate their potential for contact transfer.

## 3.2. The contact transfer process

A significant portion of this study was devoted to optimizing the conditions for creating the spills and performing the contact transfer experiments in a reproducible manner. The angle of approach of the simulated finger to the substrate and the applied pressure during the contact between the surfaces under test are known to be of critical importance to the contact transfer efficiency for powders [32]. Preliminary research from our group has also indicated that an applied shear force is efficient for transferring nanomaterials onto secondary substrates [33]. Shear contact is challenging to reproducibly mimic in the laboratory due to technical difficulties with sliding of the substrates, the need to control

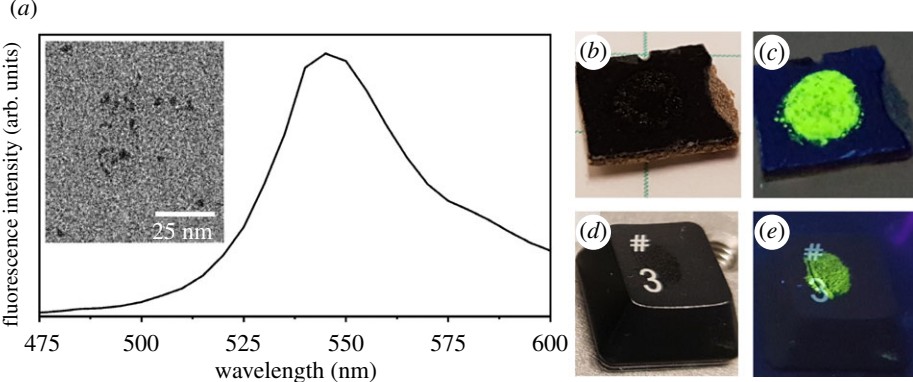

**Figure 2.** Characterization of the CdSe QDs: (*a*) fluorescence emission spectrum of the QDs suspended in toluene with an excitation wavelength of 400 nm. The inset in (*a*) depicts a typical TEM image of the QDs. (*b,c*) A series of images demonstrating the presence of residue containing QDs that was transferred to a pristine section of countertop as observed under (*b*) white light illumination and (*c*) UV illumination at 254 nm. This section of countertop was approximately 1 cm². (*d,e*) Images of a key from a computer keyboard with residue containing the QDs, which was transferred following a secondary contact with the gloved finger. These images correspond to the key observed using (*d*) white light illumination and (*e*) UV illumination at 254 nm.

both the normal and parallel forces, and complications arising from inhomogeneities in the glove and the substrates (e.g. roughness of each). The experiments in the study reported herein simulated contact under a normal load with a vertical movement of the simulated finger, as shown in the set-up in figure 1*b*. The movement of the simulated finger was adjusted to a load of 475 g during contact (4.6 N) to achieve an applied pressure of approximately 2 kg cm⁻² and a contact area between the gloved finger and the substrate of 0.23 cm² (electronic supplementary material, figure S1). The pressure used is similar to firmly pressing a button with the index finger. The use of an electromechanical device was favoured in comparison with performing the tests using human hands for both the safety of the researchers involved in this study and the reproducibility of the experimental results. The electronic, motorized control of the movement of the simulated finger enabled a high reproducibility of the load applied during contact, with a standard deviation of less than 3% ($n = 39$) in the load values.

## 3.3. Controlling the amount and drying time of the simulated spills

The amount of material deposited in the simulated spills could influence the potential for transfer of material and the variability between analyses. Simulated spills containing QDs were prepared on pristine sections of CT using either 3, 6 or 9 µl of a toluene suspension of the QDs. These spills correspond to 3.6, 7.2 and 10.8 µg of Cd, respectively, as determined by ICP-MS. These amounts were selected as representative of small splashes that can be generated during the handling of liquid QD suspensions. Each suspension was drop cast onto the centre of a section of CT and dried over a period of 24 h. These primary dry spills were each contacted with the simulated finger, covered with a pristine nitrile glove, to evaluate if the primary spills and secondary transfers were visible. Images in figure 3*a* show three sequential secondary transfers from the gloved finger to a series of CTs. The primary spills were each created using the volumes of QD suspension indicated to the left of each series of images. No clear visual differences were observed under UV illumination with regard to the fluorescence intensity of the transferred residue resulting from the secondary contact with the gloved finger. These results also indicated that spills as small as 3 µl, containing 3.6 µg of Cd for our specific suspension of QDs, could be propagated from a primary spill to multiple subsequent surfaces through the processes of contact transfer. The mid-point of this range of primary spill volumes, 6 µl, was selected for further study. Larger volumes were difficult to work with as the simulated, primary spills spread towards the edges of the sections of the CT. As a result, there was a residue from these larger spills that might not be in contact with the gloved finger during the subsequent experiments (i.e. simulated contact).

The influence of the drying time of the simulated spills was also tested for its potential impact on the processes of contact transfer. Three different time intervals for drying the spills were explored to simulate the conditions under which spills would dry in the workplace. Spills prepared with 6 µl (mid-point of the volume range of the previous study) of the suspension of QDs were drop cast onto clean, pristine sections

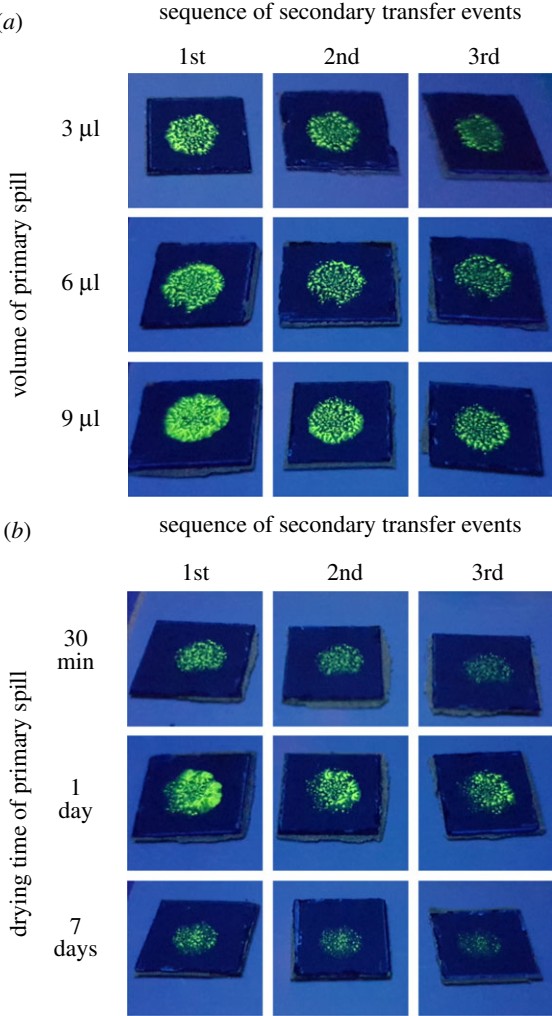

**Figure 3.** Images obtained under UV illumination at 254 nm of the residue transferred to a series of countertops by secondary contact. Each series resulted from a secondary contact with a different gloved finger following an initial contact with a primary spill. Primary spills were prepared on pristine sections of the countertop using either (a) different volumes of the suspension of QDs that were dried for 1 day before initiating contact with a gloved finger or (b) 6 µl of the suspension of QDs that were dried for 30 min, 1 day or 7 days before initiating contact with the spill.

of CT and left to dry under ambient conditions for 30 min, 24 h or 7 days. This range of times was selected to represent the contact with a small spill soon after or many days after handling a solution of QDs. Images in figure 3b show that, even after 7 days, a visible amount of residue containing the QDs is transferred by contact with a pristine nitrile glove and subsequent secondary contact with a series of pristine sections of CTs. The amount of material transferred after 7 days appeared to be less than that dried for shorter period of time. This experiment does, however, indicate that both fresh and 7-day-old dry spills can be propagated.

## 3.4. Evaluating the potential for contact transfer to different substrates

The preliminary studies on the volume of solution used to prepare the primary spills and the drying time of these spills were utilized to guide the design of the subsequent studies. A quantity of 6 µl (corresponding to $7.2 \pm 1.8$ µg of Cd) contained approximately $3 \times 10^{13}$ nanoparticles. The drying time for these spills was set to 30 min for the remaining experiments. The drying time of 30 min simulated contact with spills resulting from activities performed by workers during a single workday. Under these test conditions, the nominal area of the primary spill was $0.22 \pm 0.02$ cm$^2$ based on the analysis of 12 independent samples by fluorescence imaging (see electronic supplementary material, Section S1 and figure S1). This area was similar to the contact area of 0.23 cm$^2$ achieved between the simulated finger and the CT at a load of 475 g. This approach to the experimental design ensured that the area

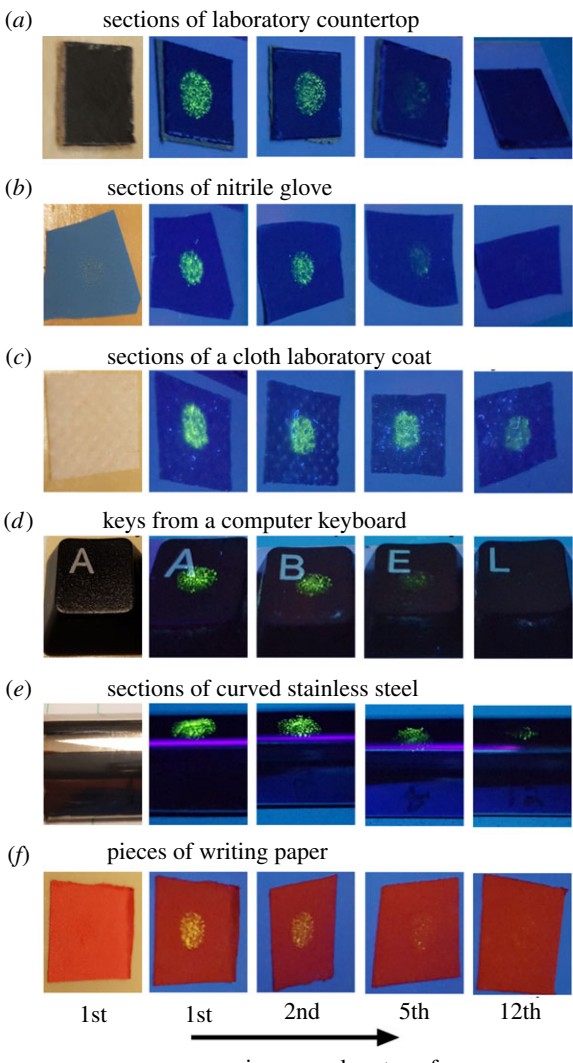

(a) sections of laboratory countertop

(b) sections of nitrile glove

(c) sections of a cloth laboratory coat

(d) keys from a computer keyboard

(e) sections of curved stainless steel

(f) pieces of writing paper

1st    1st    2nd    5th    12th

successive secondary transfers

**Figure 4.** Images of QD residue transferred by secondary contact of a gloved finger with different substrates. Separate primary spills for each series were prepared from 6 µl of the QD solution cast and dried for 30 min on a pristine section of countertop before contact with a gloved finger. Images in the first column are substrates observed by white light illumination after the first secondary contact with the gloved finger. Subsequent columns show a selection of the substrates as observed under UV illumination following a series of secondary contact events.

covered by the simulated spill was in contact with the gloved finger during the subsequent tests for material transfer.

A series of primary spills were created from 6 µl of the QD suspension with a drying time 30 min before initiating contact. During contact between a pristine nitrile glove on the simulated finger and the primary spill, the applied vertical load was maintained at 475 g for 1 s. The same load was applied for the secondary transfers between the gloved finger and the pristine substrates under test. The potential for secondary transfer was evaluated using a range of different substrates. Material transfer to the substrates was initially assessed by fluorescence imaging while under UV illumination (figure 4). A full series of images are provided in electronic supplementary material, figure S2. The fluorescence intensity did not decrease significantly until reaching the 8th to the 12th secondary transfer. The fluorescence intensity observed for the associated residue containing QDs appeared to be correlated with each type of substrate. The amount of transferred material is probably influenced by the composition, roughness and hydrophobicity of the secondary substrate. Quantifying the roughness and hydrophobicity of these diverse substrates is outside the scope of this work. The differences in roughness between a laboratory coat and a polished piece of metal are, however, easy to notice. The optical properties of the substrates are also quite diverse. These differences influenced our ability to observe the QDs on each substrate by visual inspection using fluorescence imaging techniques. For

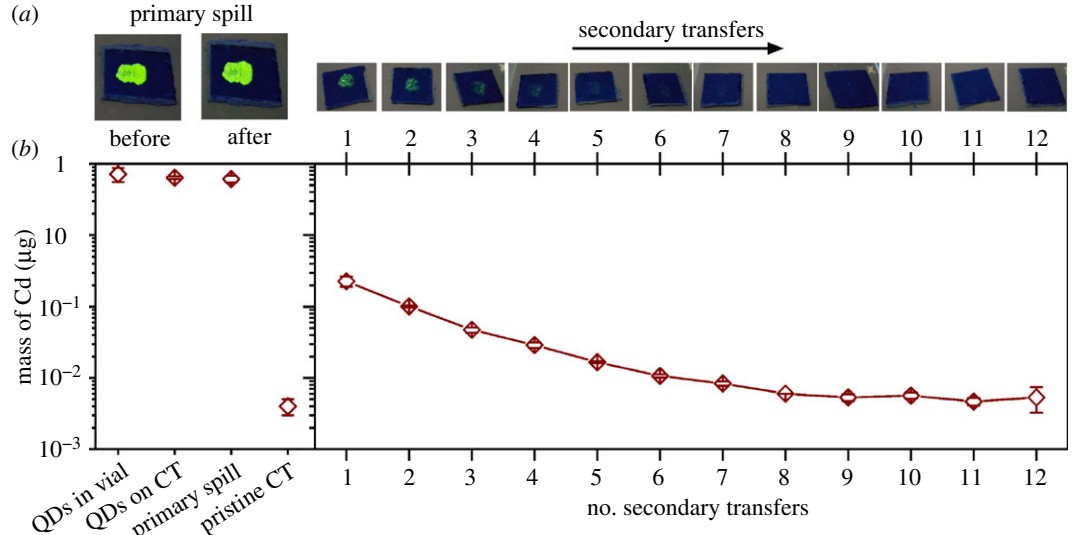

**Figure 5.** Results from elemental analyses of the residue transferred as a result of contact of a gloved finger with a primary spill of QDs and subsequent contact with a series of pristine sections of countertop (CT), simulating secondary transfer events. (*a*) Optical images obtained under UV illumination showing the primary spill before and after contact with the gloved finger (left), and the secondary transfer of residue containing the QDs from the glove to 12 separate sections of CT (right). (*b*) The amount of Cd present in each of these samples, and a couple of control samples, as determined by ICP-MS. The error bars represent one standard deviation from the mean values calculated from several independent measurements (*n* = 3, except for pristine CT where *n* = 6). Error bars for most of the samples are proportional to or smaller than the symbol used in the plot.

example, the QDs are easily observed on the sections of the laboratory coat, which is attributed to the ability of this substrate to reflect the incident UV light. In contrast, many of the other substrates absorb more strongly in the UV range. Nevertheless, the presence of residue containing QDs was detected for all tested substrates under UV illumination, and it could go undetected without the use of specific tools (e.g. UV illumination) or techniques to assist in monitoring for their presence.

A quantitative analysis of the secondary spill images like the one performed by Clemente *et al.* [13] is beyond the scope of the current work. This image analysis approach would not be feasible for the direct evaluation of workplace surfaces due to the variety of optical properties of potential substrates. It is important to note that the residue from the QDs after secondary transfer to most of the substrates was not easily observed under white light illumination. The analysis of each substrate by fluorescence imaging was necessary to assess the potential for material transfer. The results of the analyses by fluorescence imaging indicate that, under the conditions tested in this study, the secondary transfer of nanomaterials from the gloved finger occurred for each of the substrates evaluated in this study. The amount of material transfer was quantified through ICP-MS only for secondary transfer to pristine sections of the CT.

## 3.5. Quantitative analysis of material transfer from/to countertops

Inductively coupled plasma mass spectrometry (ICP-MS) techniques were used to quantify the amount of QDs transferred through secondary contact to pristine sections of the CT. A series of solutions with 2% (v/v) $HNO_3$ and measured amounts of dissolved Cd species (e.g. 0.1, 0.25, 1, 2.5, 5 and 10 ppb) were measured by ICP-MS to obtain a calibration curve for the experimental samples (electronic supplementary material, figure S3). Pristine glass vials, identical to the ones used for sample digestion, and six sections of pristine CT that had been in direct contact with pristine nitrile gloves, were analysed to evaluate background levels of cadmium. These pristine sections of CT and nitrile gloves were brought into contact using the same mechanical controls and protocol for contact used to evaluate material transfer. Simulated spills containing the QDs were analysed by drop casting 6 µl of the suspension of QDs directly into pristine glass vials (referred to as 'QDs in vial' in figure 5 and electronic supplementary material, table S1). The same amount of QDs was also drop cast onto sections of pristine CT and dried for 24 h before ICP-MS analysis (referred to as 'QDs on CT' in figure 5 and electronic supplementary material, table S1). The amount of cadmium detected from the

analysis of the pristine vials and pristine sections of CT was negligible when compared with the amounts found in these samples containing 6 µl aliquots taken from the solution of QDs (electronic supplementary material, table S1). The amount of cadmium detected from the solution of QDs directly spotted into glass vials was, within experimental error, similar to the amounts detected from the solution of QDs spotted on the sections of the CT. This indicates good transfer of QDs from solution to the CT, and a good recovery of QDs on CTs through the acid digestion process. The average amount of Cd was calculated for each of the samples from the replicate measurements. The variation in these measurements is reported as one standard deviation from these mean values. The concentration of Cd in the original suspension of QDs was determined to be $1.2 \pm 0.3$ mg ml$^{-1}$. The total amount of Cd in the simulated spills created from 6 µl of this solution was estimated to be $7.2 \pm 1.8$ µg ($n = 3$). These results indicate that the methods outlined herein enabled us to reproducibly handle, transfer and analyse the suspensions of QDs. In addition, for these amounts of Cd, the use of ICP-MS will be a reliable technique for quantitatively assessing the reproducibility of the methods and for monitoring the amount of material in the primary spills and the amount of material transferred through secondary contact events. The methods of digestion and dilution can be adapted for settings where higher or lower amounts of Cd might be expected in further studies.

The techniques established for sample preparation and analysis by ICP-MS were applied to a triplicate series of contact transfers from primary spills to pristine sections of the CT. Each set of samples analysed by ICP-MS consisted of a primary spill after contact with a pristine glove on the simulated finger and 12 additional sections of CT from the subsequent secondary contact transfer events (figure 5). The volume of QD suspension used to prepare the primary spills was 6 µl for this series of studies. The simulated spills were dried for 30 min before initiating the contact transfer experiments.

The amount of cadmium transferred from a gloved finger to a series of pristine sections of CT progressively decreases through the successive steps of contact transfer (figure 5). The first substrate of the series of substrates prepared by secondary transfer contained the highest amount of Cd at $0.23 \pm 0.04$ µg. Based on the average amount of Cd that remained in the primary spill after contact with the gloved finger ($6.1 \pm 0.6$ µg), and the amount of Cd in the initial spill before contact ($7.2 \pm 1.8$ µg within the 6 µl used to create the initial spill), up to 1.1 µg of the Cd from the residue in the primary spill was transferred to the gloved finger following a single contact with the spill.

The largest amount of Cd transferred to the sections of the CT took place in the first secondary contact, as expected, accounting for $0.23 \pm 0.04$ µg. The amount of Cd transferred to each subsequent section of CT decreases from the previous secondary transfer event. It is likely that Cd is still transferred in small amounts beyond the ninth step of the simulated secondary contact events, but the signal from these sections of CT was indistinguishable from that of the pristine sections of CT using the methods reported herein. The cumulative amount of Cd detected on the CTs across all 12 secondary transfers was $0.47 \pm 0.05$ µg. Therefore, after 12 secondary contact transfers, it is estimated that up to 0.63 µg of Cd remained on the glove.

The amount of QDs transferred to the pristine sections of the CT after eight sequential contact transfer events was indistinguishable from the amount found on pristine sections of the CT. Images in figure 5a show representative optical images obtained under UV illumination from one series of the analysed sections of the CT. The QDs were not detectable in the fluorescence images after the seventh secondary transfer event, indicating that ICP-MS was more sensitive for QD detection on CTs than UV visual inspection. The standard deviation for the amount of Cd detected in the residue from the first of the secondary transfer events was 16% of the absolute value, and the standard deviation associated with each of the subsequent secondary transfer events was less than 10%. These results again show the reproducibility for the processes of contact transfer using the described set-up and the conditions evaluated herein for material transfer.

The quantity of Cd per sample as determined by ICP-MS can be used to roughly estimate the number of QDs present. The QDs were assumed to contain a 1 : 1 atomic ratio of cadmium and selenium, to have a CdSe mass density of 5.81 g ml$^{-1}$ [34], and to be nominally spherical particles with an average diameter of 5 nm (i.e. a value of 26.2 nm$^3$ or 0.0015 ng per particle). Under these assumptions, the original spill contains approximately $3 \times 10^{13}$ nanoparticles or around 30 trillion nanoparticles. Approximately $10^{12}$ nanoparticles are transferred upon the first contact of the residue on the glove with a pristine section of CT, and this amount of nanoparticles already goes unnoticed on many substrates by unaided visual inspection. In addition, the amount of particles transferred in the first instance of secondary contact is below the detection limits of X-ray fluorescence (XRF) spectroscopy or scanning electron microscopy (SEM)-based energy-dispersive X-ray spectroscopy (EDX) techniques (see electronic supplementary material, figures S4 and S5). Handheld XRF devices could become commonplace for heavy metal

detection in the workplace, as they are portable and fast, and are already being used for example in mining operations for the evaluation of soils and ores. A technique that is popular in research laboratories and industries that focused on nanotechnology, i.e. SEM-EDX, could be used by Environmental Health and Safety (EHS) safety officers or workers to evaluate contamination from engineered nanomaterials in the workplace by analysing swabs. However, our preliminary assessment of these techniques in comparison with UV-assisted inspection or ICP-MS analyses indicates that XRF and SEM-EDX are not as sensitive as these other techniques towards the CdSe QDs.

## 3.6. Occupational health and safety significance

The results of these analyses demonstrate that a spill of a few micrograms of CdSe QDs or trillions of nanoparticles can be unintentionally spread by a single contact with a glove and the subsequent contact of that contaminated glove with pristine sections of CT or other surfaces. The transfer of Cd from contaminated to supposedly clean areas can take place by contact if proper hygiene procedures are not followed in the workplace (e.g. frequent changing of gloves, wiping up small spills or residue from splashes). The transfer could even take place from contact with a 7-day-old spill on a CT. Contact transfer of residue containing QDs could lead to a chronic exposure to workers. Exposure routes could include the uptake of cadmium through dermal absorption or the inadvertent ingestion of residue containing Cd. While workers might observe safe work practices in areas where Cd contamination is expected, they might not wear the appropriate personal protective equipment in other areas that are deemed to be clean and they may be unaware of contact with residue, such as that containing the CdSe QDs, which cannot be observed with the unaided eye (e.g. without UV illumination or elemental analysis).

Workers could be surpassing the recommended daily intake limits of Cd if they were chronically exposed to Cd in the workplace, or if large amounts of CdSe QDs were handled and spilled with inadvertent exposure. In this study, a detectable fraction of cadmium was transferred in the form of CdSe QDs to multiple substrates, as demonstrated by the fluorescence images of the residue on these surfaces. Contact transfer of CdSe QDs within the workplace is, therefore, a newly demonstrated mode of occupational exposure to Cd, which could lead to a chronic exposure above maximum recommended levels.

The unintentional transfer of engineered nanomaterial residue between surfaces commonly found in the workplace is important when evaluating the potential of workers' exposure to these materials. We have shown that gloves can contain and transport CdSe QDs after a touch event with a single spill. The amount of CdSe QDs found in the residue from secondary transfer events could lead to an accumulation of QDs on surfaces throughout the workplace, as well as an unnoticed systemic exposure to workers. This possible route for exposure should be considered in future work for guidelines and policies on workplace hygiene and glove use. The knowledge from these studies and the techniques developed herein could also help with the development of acceptable surface limits (ASLs) for engineered nanomaterials. ASLs are the amount of a chemical or material found on workplace surfaces that are considered to be at a safe level for workers to be exposed to via dermal contact [35], The ASLs still need to be defined for engineered nanomaterials, including CdSe QDs. The studies demonstrated herein could potentially impact occupational exposure limits and best practices for workers when preparing and handling engineered nanomaterials.

In addition to the concerns in the workplace, there remains the concern for proper disposal of materials that could result in potential contact with engineered nanomaterials, such as the CdSe QDs. Up to 1.1 µg of Cd was transferred to the nitrile glove following a primary contact with residue spilled on sections of the CT. From the transferred residue, up to 0.63 µg of Cd could remain on the glove even after 12 consecutive secondary transfer events. These results should be considered when developing safety protocols for handling waste that has been in contact with engineered nanomaterials. The proper disposal of personal protective equipment, and other materials, will be necessary to avoid the release of Cd into the environment. Cadmium selenide QDs have been shown to have toxic effects on biological systems [36], and other engineered nanomaterials are also under scrutiny. Appropriate disposal of personal protective equipment and other materials will be necessary to avoid introducing Cd into ecosystems.

It is likely that other engineered nanomaterials could be transferred through similar mechanisms of contact transfer. A comparable amount of residue could also be transferred to surfaces throughout the workplace, and potentially beyond. Specific types of engineered nanomaterials and other relevant workplace surfaces can be evaluated using the protocols established in this study. The properties of the surfaces under study must be considered when evaluating the presence of engineered nanomaterials. For example, if using elemental analysis for quantification of iron oxide nanoparticles

on steel surfaces, a significant background signal from the iron in steel can result in increased detection limits. If the engineered nanomaterials are not fluorescent, additional techniques specific to the properties of the target engineered nanomaterials could be used. Non-fluorescent engineered nanomaterials can be tagged with fluorescent species for tracking purposes, as suggested by Clemente *et al.* [13]. Work practices and risk assessment models may need to be revised in the light of the contact transfer of engineered nanomaterials. Some potential recommendations based on this study include an increased frequency of surface cleaning and proper waste segregation for personal protective equipment and disposables that have been in contact with engineered nanomaterials. The probability for secondary transfer of engineered nanomaterials must be evaluated for other engineered nanomaterials, including assessing the extent of their contact transfer. These factors should be included in risk assessment models, as well as in the training of employers, workers and regulators.

# 4. Conclusion

Herein, we studied the potential for cadmium selenide QDs with a hydrophobic coating to undergo transfer between surfaces typically found in the workplace as a result of simulated contact by a worker. It was determined that residue containing these nanomaterials will transfer from nitrile gloves upon contact with pristine substrates. A mechanical set-up was built to reproducibly simulate contact of a gloved finger with a section of CT containing spilled residue and subsequently to contact a series of other substrates. The residue transferred to a gloved finger by contact with a simulated spill on a section of laboratory CT can be transferred to other sections of the CT, to a laboratory coat, to a computer keyboard, to another glove, to writing paper, or to a door handle. A single contact between the gloved finger and a simulated spill prepared from approximately 6 μl of a solution containing these QDs could be transferred through successive contacts with at least 12 substrates. The transfer could still take place even for a 7-day-old spill. This study could be extended in the future to evaluate additional conditions that represent other aspects of the typical workplace environments and workflows encountered daily in laboratories. We have provided here a detailed description of the experimental set-up and analytical methods that will enable the extension of this study to evaluate other substrates, as well as nanomaterials of other compositions, sizes, shapes and surface chemistries. Additional studies of interest include contact by other materials, for example simulating placing a glass beaker on top of a spill, evaluating other applied pressures and contact times, propagating engineered nanomaterials from spills of dry powders (as opposed to dried spills pursued herein that were created by solvent evaporation) or from nanomaterials capped with a hydrophilic coating, and extending the work to shear contact to simulate other worker habits. The potential for the transfer of engineered nanomaterials from the accidental contact of a worker with a small amount of residue in the workplace should be considered when developing risk assessment programmes in research laboratories and industries working with engineered nanomaterials.

Data accessibility. Electronic supplementary material is available: further details for the experimental methods used to prepare, handle and characterize the materials used in this study, additional results from the elemental analyses of these materials (e.g. additional ICP-MS data, as well as EDX and XRF data), further characterization of the applied load and contact area, and additional SEM, optical and fluorescence images of the samples. The data are provided in electronic supplementary material [37].

Authors' contributions. I.A. designed experiments, synthesized and characterized nanoparticles, performed digestions for elemental analysis and wrote manuscript; T.M.N. designed experiments, synthesized nanoparticles, selected and prepared substrates, performed contact transfer experiments and photography and edited manuscript; V.P. designed experiments, guided synthesis and purification of nanoparticles, designed methods for elemental analysis, performed digestions for elemental analysis, ran elemental analysis and edited manuscript; M.W.B. characterized nanoparticles and edited manuscript; K.E.C.C. assisted in initial design of experiments, design of methods for elemental analysis, literature analysis and edited manuscript; M.T.Y.P. assisted with conceptualization of the project, microscopy studies and edited manuscript; T.C.H.C. synthesized nanoparticles and edited manuscript; C.B.D. performed digestions for elemental analysis and edited manuscript; B.D.G. conceived of the study, designed experiments, raised funding, oversaw the project and edited manuscript. All authors assisted in revisions to the manuscript and evaluation of the data, and all authors gave final approval for publication and agree to be held accountable for the work performed therein.

Competing interests. The authors declare no competing financial interest.
Funding. This research was supported in part by WorkSafeBC (Workers' Compensation Board of British Columbia) through the Innovation at Work research programme (Project no. RS2014-SP04), the Natural Sciences and Engineering Research Council of Canada (NSERC; grant no. RGPIN-2020-06522), the Canada Research Chairs Program (B.D.G., grant no. 950-215846), the Simon Fraser University Work Study Program (T.M.N.), and CMC

Microsystems (MNT grant no. 3977). This work made use of 4D LABS shared facilities supported by the Canada Foundation for Innovation (CFI), British Columbia Knowledge Development Fund (BCKDF), Western Economic Diversification Canada and Simon Fraser University.

Acknowledgements. We would like to acknowledge the contribution of Dr Stefano Rubino for preliminary work leading to this project. We also thank Jeffrey Walker and Shayne Harrel of Olympus, as well as Nicola Struyk, Christopher Butz and Scott Curry of Innov-X Canada for ongoing support of this research project through in-kind contributions of their time and equipment. We also thank the contributions of the additional members of the Steering Committee for this project: Dr Mark Teo, Senior Policy Advisor with WorkSafeBC in the Policy, Regulation and Research Division; Ms. Catherine Peltier, Director, Research and Laboratory Safety for Environmental Health and Safety at Simon Fraser University and Mr Mike Neudorf, former Director with Environmental Health & Safety at Simon Fraser University.

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
