## [Peer Review File · Royal Society Open Science]

Review History

RSOS-210141.R0 (Original submission)

Review form: Reviewer 1

Is the manuscript scientifically sound in its present form?

No

Are the interpretations and conclusions justified by the results?

Yes

Is the language acceptable?

Yes

Do you have any ethical concerns with this paper?

No

Have you any concerns about statistical analyses in this paper?

No

Recommendation?

Major revision is needed (please make suggestions in comments)

Comments to the Author(s)

I don't find this manuscript as scientifically very interesting for publication. After carefully reading the whole manuscript I feel the authors have elaborated lot of thing unnecessarily. Since the target of this work is to estimate the extent of contact transfer of engineered nanomaterials to make a suitable protocol for their safe handling in industries and regulatory agencies, authors should stick primarily to fluorescence based technique (ex. for CdSe) as other techniques are either too expensive or too specialized for any commercial agency. Moreover, since they are not synthesizing any new materials and all the characterization for CdSe is already known to us, authors may avoid material characterization part. I strongly feel that this is a publishable material but needs a major revision as I mentioned previously.

Besides these serious points, other noted observations are:

- (i) Page 3, L-4, remove today
- (ii) Page 3, L-6-7: Estimations show that in 2020, approximately 6 million workers worldwide would be handling engineered nanomaterials. This is already 2021. Author should modify these lines with current statistics.
- (iii) Is the surface's hydrophobic nature is a controlling factor for effective transfer? If so then how to avoid or normalize this factor? Due to their smaller size, air flow and humidity could also be other serious factors.
- (iv) Why there is a potential interference between the signal from the selenium species and the argon carrier gas used in the ICP-MS analyses? Since the atomic mass of Se and Ar are 78.96 and 39.95, there should not be any interference between them unless Ar makes Ar₂ cluster in the plasma. Since the atomic mass of Cd is 112.41, in principle both Cd and Se can be detected simultaneously without any interference.
- (v) The section starts with "Cadmium is known to affect renal function and to....", in Page#20, should go to the Introduction section.
- (vi) Conclusion is too big.

Review form: Reviewer 2

Is the manuscript scientifically sound in its present form?

Yes

Are the interpretations and conclusions justified by the results?

Yes

Is the language acceptable?

Yes

Do you have any ethical concerns with this paper?

No

Have you any concerns about statistical analyses in this paper?

No

Recommendation?

Accept with minor revision (please list in comments)

Comments to the Author(s)

This is a very well written manuscript and I have few specific comments. One general comment I have is if the authors gave any considerations to replicating the experiments with the dry powder material as opposed to the material in solution? I realize they allowed the solution to dry, but for many powders in solution are less dispersible even after fully drying.

Specific comments

- Page 18 Line 49 – It is stated that it is likely that Cd is transferred beyond the 9th step, but it was below the LOD for ICP-MS. How were the datapoints in Figure 5 determined past the 9th secondary transfer if they were below the LOD? This should be clarified in the text.
- Page 21 Line 49 – “This possible route for exposure should be considered in future work for guidelines and policies on workplace hygiene.” I would consider adding “and glove use” so that way it reads “policies on workplace hygiene and glove use”. I think a key concept you are fully considering is that personal protective equipment, such as gloves, if properly used and regularly changed when work is completed with nanomaterials, should limit the amount of transfer events and spread around the workplace. I think this should be emphasized here. If a worker disposes of the dirty gloves and dons a new pair, it would considerably reduce the amount of material transfer.

Decision letter (RSOS-210141.R0)

Dear Professor Gates:

Title: Contact Transfer of Engineered Nanomaterials in the Workplace
Manuscript ID: RSOS-210141

The editor assigned to your manuscript has now received comments from reviewers. We would like you to revise your paper in accordance with the referee and Subject Editor suggestions which can be found below (not including confidential reports to the Editor). Please note this decision does not guarantee eventual acceptance.

Please submit your revised paper before 07-May-2021. Please note that the revision deadline will expire at 00.00am on this date. If we do not hear from you within this time then it will be assumed that the paper has been withdrawn. In exceptional circumstances, extensions may be possible if agreed with the Editorial Office in advance. We do not allow multiple rounds of revision so we urge you to make every effort to fully address all of the comments at this stage. If deemed necessary by the Editors, your manuscript will be sent back to one or more of the original reviewers for assessment. If the original reviewers are not available we may invite new reviewers.

On behalf of the Subject Editor Professor Anthony Stace and the Associate Editor Professor Chaohua Cui.

RSC Associate Editor:
Comments to the Author:
(There are no comments.)

RSC Subject Editor:
Comments to the Author:
(There are no comments.)

Reviewers' Comments to Author:
Reviewer: 1

Comments to the Author(s)

I don't find this manuscript as scientifically very interesting for publication. After carefully reading the whole manuscript I feel the authors have elaborated lot of thing unnecessarily. Since the target of this work is to estimate the extent of contact transfer of engineered nanomaterials to make a suitable protocol for their safe handling in industries and regulatory agencies, authors should stick primarily to fluorescence based technique (ex. for CdSe) as other techniques are either too expensive or too specialized for any commercial agency. Moreover, since they are not synthesizing any new materials and all the characterization for CdSe is already known to us, authors may avoid material characterization part. I strongly feel that this is a publishable material but needs a major revision as I mentioned previously.

Besides these serious points, other noted observations are:

- (i) Page 3, L-4, remove today

- (ii) Page 3, L-6-7: Estimations show that in 2020, approximately 6 million workers worldwide would be handling engineered nanomaterials. This is already 2021. Author should modify these lines with current statistics.
- (iii) Is the surface's hydrophobic nature is a controlling factor for effective transfer? If so then how to avoid or normalize this factor? Due to their smaller size, air flow and humidity could also be other serious factors.
- (iv) Why there is a potential interference between the signal from the selenium species and the argon carrier gas used in the ICP-MS analyses? Since the atomic mass of Se and Ar are 78.96 and 39.95, there should not be any interference between them unless Ar makes Ar₂ cluster in the plasma. Since the atomic mass of Cd is 112.41, in principle both Cd and Se can be detected simultaneously without any interference.
- (v) The section starts with "Cadmium is known to affect renal function and to....", in Page#20, should go to the Introduction section.
- (vi) Conclusion is too big.

Reviewer: 2

Comments to the Author(s)

This is a very well written manuscript and I have few specific comments. One general comment I have is if the authors gave any considerations to replicating the experiments with the dry powder material as opposed to the material in solution? I realize they allowed the solution to dry, but for many powders in solution are less dispersible even after fully drying.

Specific comments

- Page 18 Line 49 – It is stated that it is likely that Cd is transferred beyond the 9th step, but it was below the LOD for ICP-MS. How were the datapoints in Figure 5 determined past the 9th secondary transfer if they were below the LOD? This should be clarified in the text.

- Page 21 Line 49 – "This possible route for exposure should be considered in future work for guidelines and policies on workplace hygiene." I would consider adding "and glove use" so that way it reads "policies on workplace hygiene and glove use". I think a key concept you are fully considering is that personal protective equipment, such as gloves, if properly used and regularly changed when work is completed with nanomaterials, should limit the amount of transfer events and spread around the workplace. I think this should be emphasized here. If a worker disposes of the dirty gloves and dons a new pair, it would considerably reduce the amount of material transfer.

Author's Response to Decision Letter for (RSOS-210141.R0)

See Appendix A.

RSOS-210141.R1 (Revision)

Review form: Reviewer 1

Is the manuscript scientifically sound in its present form?

Yes

Are the interpretations and conclusions justified by the results?

Yes

Is the language acceptable?

Yes

Do you have any ethical concerns with this paper?

No

Have you any concerns about statistical analyses in this paper?

No

Recommendation?

Accept as is

Comments to the Author(s)

The authors have corrected all the raised questions and justified all the queries. Hereby I am recommending publishing it as is.

Review form: Reviewer 2

Is the manuscript scientifically sound in its present form?

Yes

Are the interpretations and conclusions justified by the results?

Yes

Is the language acceptable?

Yes

Do you have any ethical concerns with this paper?

No

Have you any concerns about statistical analyses in this paper?

No

Recommendation?

Accept as is

Comments to the Author(s)

No additional comments.

Decision letter (RSOS-210141.R1)

Dear Professor Gates:

Title: Contact Transfer of Engineered Nanomaterials in the Workplace
Manuscript ID: RSOS-210141.R1

It is a pleasure to accept your manuscript in its current form for publication in Royal Society Open Science. The chemistry content of Royal Society Open Science is published in collaboration with the Royal Society of Chemistry.

On behalf of the Subject Editor Professor Anthony Stace and the Associate Editor Professor Chaohua Cui.

RSC Associate Editor:
Comments to the Author:
(There are no comments.)

RSC Subject Editor:
Comments to the Author:
(There are no comments.)

Reviewer(s)' Comments to Author:
Reviewer: 2

Comments to the Author(s)
No additional comments.

Reviewer: 1

Comments to the Author(s)

The authors have corrected all the raised questions and justified all the queries. Hereby I am recommending publishing it as is.

Appendix A

Byron D. Gates
Professor, Department of Chemistry
Associate Member, Mechatronic Systems Engineering
Simon Fraser University, 8888 University Dr.
Burnaby, B.C., Canada, V5A 1S6

Phone: (778)-782-8066
Fax: (778)-782-3765
Email: bgates@sfu.ca

May 20, 2021

Prof. Anthony Stace, Subject Editor

Prof. Chaohua Cui, Associate Editor

Dear Professors Stace and Cui,

I enclose our revised manuscript titled “Contact Transfer of Engineered Nanomaterials in the Workplace” authored by Irene Andreu, Tuan M. Ngo, Viridiana Perez, Matthew W. Bilton, Kelly E.C. Cadieux, Michael T.Y. Paul, Tania C. Hidalgo Castillo, Clifton Bright Davies, Byron D. Gates (Manuscript ID: RSOS-210141) for your further consideration as a research article in *Royal Society Open Science*.

The referees have raised important points. We have carefully addressed their concerns with revisions to the manuscript. We also include below our replies to each of their concerns along with the new or revised text (both highlighted in red) that were incorporated into the revised manuscript. The revised sections are also highlighted in red in one of the copies of the submitted manuscript. We believe that our revised manuscript has been strengthened and clarified by the feedback from the referees. We also believe that our findings are an optimal fit to *Royal Society Open Science* as outlined in our cover letter.

I confirm that this manuscript is an original study. The contents are not under consideration by any other journal.

Sincerely,

Byron Gates

Reviewers' Comments to Author and Our Replies and Revisions (highlighted in red):

Reviewer: 1

I don't find this manuscript as scientifically very interesting for publication. After carefully reading the whole manuscript I feel the authors have elaborated lot of thing unnecessarily. Since the target of this work is to estimate the extent of contact transfer of engineered nanomaterials to make a suitable protocol for their safe handling in industries and regulatory agencies, authors should stick primarily to fluorescence based technique (ex. for CdSe) as other techniques are either too expensive or too specialized for any commercial agency.

We appreciate the time and comments from the reviewer. We agree that fluorescence-based techniques are more suitable than ICP-MS for point-of-use settings when the nanomaterials exhibit a unique fluorescence signature to other materials in the workplace. The goal of the comparison between fluorescence assessment and quantitative analysis via ICP-MS, XRF or other alternative techniques is to provide guidance on the detection limits of the fluorescence techniques and to exemplify how our proposed framework can be implemented for other engineered nanomaterials.

Moreover, since they are not synthesizing any new materials and all the characterization for CdSe is already known to us, authors may avoid material characterization part.

We agree with the reviewer that CdSe QD synthesis is not the focus of this manuscript. Characterization was included to provide insights into the properties of the materials that were prepared and used in these studies as changes to the synthetic conditions would yield distinct materials (e.g., resulting in a shift in position of the peak emission intensity). Our discussion of these materials is relatively minor (e.g., one paragraph in the manuscript). We believe the inclusion of these details provides context for the discussion.

I strongly feel that this is a publishable material but needs a major revision as I mentioned previously.

Besides these serious points, other noted observations are:

(i) Page 3, L-4, remove today

We have performed the suggested change.

(ii) Page 3, L-6-7: Estimations show that in 2020, approximately 6 million workers worldwide would be handling engineered nanomaterials. This is already 2021. Author should modify these lines with current statistics.

We have searched multiple sources and have not located an updated estimation to this prediction. Likely there have been delays in new estimates due in part to the pandemic. We have revised our statement to read:

“One estimation suggested that there could be as many as 6 million workers worldwide who handled engineered nanomaterials in 2020.”

(iii) Is the surface's hydrophobic nature a controlling factor for effective transfer? If so then how to avoid or normalize this factor?

We agree with the reviewer that there are many factors influencing the contact transfer of nanomaterials. The hydrophobicity of both the nanomaterial and receiving surfaces are likely important factors. For example, the interparticle cohesive forces likely are a key factor in both the contact transfer process and in the ability to clean-up a dried spill. The intermolecular interactions that form as a result of drying a spill of nanomaterials will be different for hydrophobic and hydrophilic based coatings. These coatings, their intermolecular interactions, and the interparticle interactions will be influenced by the presence of salts (e.g., from the solution), and by the temperature and humidity. We controlled for the nanoparticle properties throughout our experiments by consistently using CdSe QDs synthesized in the same way and with the same capping groups. Future studies can include testing CdSe QDs with other types of capping groups to explicitly study the influence of additional surface chemistries on the engineered nanomaterials during their contact transfer. As for the properties of the receiving surfaces, we prepared a series of representative receiving surfaces that were used as-is (i.e., without chemical treatment of their surfaces), with the goal of providing representative performances of these surfaces that are frequently found in workplace settings. We have added hydrophobicity as a factor to consider in the contact transfer efficiency in the revised manuscript:

“This study also found that the transfer efficiency is higher for powders deposited on a metallic surface than on wood, hinting at the potential influence of the substrate roughness and hydrophobicity.”

and ... “The amount of transferred material is likely influenced by the composition, roughness, and hydrophobicity of the secondary substrate. Quantifying the roughness and hydrophobicity of these diverse substrates is outside the scope of this work.”

And to the conclusion we added... “Additional studies of interest include contact by other materials, for example simulating placing a glass beaker on top of a spill, evaluating other applied pressures and contact times, propagating engineered nanomaterials from spills of dry powders (as opposed to dried spills pursued herein that were created by

solvent evaporation) or from nanomaterials capped with a hydrophilic coating, and extending the work to shear contact to simulate other worker habits.”

Due to their smaller size, air flow and humidity could also be other serious factors.

We agree with the reviewer that air flow and humidity could also affect contact transfer efficiency. We performed all experiments in the same location, protecting each sample with a closed container after their preparation and during their transport, around the same time of the day, in a low foot traffic, and in a temperature- and humidity-controlled laboratory. A new study in a controlled environmental chamber with adjusting the air flow and humidity conditions to assess their independent impacts on the results would be a great future extension of this work, but it is beyond the scope of the current manuscript. We have added a sentence to address the influence of air flow and humidity on contact transfer:

“The contact transfer experiments were performed at ambient conditions in a laboratory with minimal air flow currents and temperature or humidity fluctuations, as these could affect the contact transfer results.”

(iv) Why there is a potential interference between the signal from the selenium species and the argon carrier gas used in the ICP-MS analyses? Since the atomic mass of Se and Ar are 78.96 and 39.95, there should not be any interference between them unless Ar makes Ar₂ cluster in the plasma. Since the atomic mass of Cd is 112.41, in principle both Cd and Se can be detected simultaneously without any interference.

We understand your concern. On the surface it appears that Argon would create an ideal plasma for the analysis of selenium species. There are, however, a series of selenium isotopes that have the same atomic weights with ionized argon species [e.g., ⁸⁰Se (the isotope with the highest natural abundance for selenium) overlaps with a ⁴⁰Ar⁴⁰Ar⁺ dimer and other species of ⁴⁰Ar (the isotope with the highest natural abundance for argon)]. We have replaced the previous experimental reference to this challenge with a theoretical study that accounts for the diverse chemistry of these interferences:

G. Bouchoux, A. M. Rashad, A. I. Helal. Theoretical Investigation of Selenium Interferences in Inductively Coupled Plasma Mass Spectrometry, *J. Phys. Chem. A* 2012, **116**, 9058–9070.

(v) The section starts with “Cadmium is known to affect renal function and to....”, in Page#20, should go to the Introduction section.

We have moved the background information about Cd toxicity to the Introduction and made the appropriate changes to the revised manuscript.

“Quantum dots are often composed of toxic elements such as cadmium or lead.²⁰ Cadmium is known to affect renal function and to result in skeletal damage.^{29,30} It is also a suspected carcinogen.³¹ The Food and Agriculture Organization of the United Nations and the World Health Organization have estimated that the tolerable weekly intake of cadmium for an adult is between 400 and 500 μg .³² The toxicity of CdSe in the form of nanomaterials is still being quantified, but it is anticipated to be equal to or higher than that of micron-size powders of the same composition^{33,34} due to both the smaller dimensions and the high ratio of surface area to mass for nanomaterials relative to micron-size or larger materials.”

(vi) *Conclusion is too big.*

We have shortened the conclusion by removing some of the sentences that summarized the results. The conclusion now reads:

“Herein, we studied the potential for cadmium selenide quantum dots with a hydrophobic coating to undergo transfer between surfaces typically found in the workplace as a result of simulated contact by a worker. It was determined that residue containing these nanomaterials will transfer from nitrile gloves upon contact with pristine substrates. A mechanical set-up was built to reproducibly simulate contact of a gloved finger with a section of countertop containing spilled residue and subsequently to contact a series of **other substrates**. **The residue** transferred to a gloved finger by contact with a simulated spill on a section of laboratory countertop can be transferred to other sections of countertop, to a laboratory coat, to a computer keyboard, to another glove, to writing paper, or to a door handle. A single contact between the gloved finger and a simulated spill prepared from $\sim 6 \mu\text{L}$ of a solution containing these QDs could be transferred through successive contacts with at least 12 substrates. Transfer could still take place even for a 7-day-old spill. This study could be extended in the future to evaluate additional conditions that represent other aspects of the typical workplace environments and workflows encountered daily in laboratories. We have provided here a detailed description of the experimental set-up and analytical methods that will enable the extension of this study to evaluate other contact, as well as to nanomaterials of other compositions, sizes, shapes, and surface chemistries. Additional studies of interest include contact by other materials, for example simulating placing a glass beaker on top of a spill, evaluating other applied pressures and contact times, **propagating engineered nanomaterials from spills of dry powders (as opposed to dried spills pursued herein that were created by solvent evaporation) or from nanomaterials capped with a hydrophilic coating**, and extending the work to shear contact to simulate other worker habits. The potential for the transfer for engineered nanomaterials from the accidental contact of a worker with a small amount of residue in the workplace should be considered when developing risk assessment programs in research laboratories and industries working with engineered nanomaterials. “

Reviewer: 2

This is a very well written manuscript and I have few specific comments.

One general comment I have is if the authors gave any considerations to replicating the experiments with the dry powder material as opposed to the material in solution? I realize they allowed the solution to dry, but for many powders in solution are less dispersible even after fully drying.

We agree with the reviewer that the study of the propagation of spills prepared from dry powders is of interest. This could be a future direction of study. Dry spills are relevant in the workplace as an unnoticed source of engineered nanomaterial contact transfer. We simulated spills created from a dried solution, but spills of a dry powder would also be relevant. Wet spills might be more easily noticed and more easily cleaned up than a dry spill. Wet spills are expected to be less of a concern with regards to its persistence in the workplace if a worker cleans up these spills promptly and properly. Oversight of proper workplace hygiene would lead to a dried spill, hence our focus on this type of spill. The present study could be extended in the future to spills of dry powders but would require controlling the amount and distribution of the powder over the region of the substrate in contact with the gloved finger to achieve reproducible results. The humidity and the charge present in the dry powder will also be factors that need to be controlled in these future studies. Air flow would also be a more significant concern for spills of a dry powder than spills prepared from a dried solution. We have modified a sentence in the Conclusions to address this possible future direction.

“Additional studies of interest include contact by other materials, for example simulating placing a glass beaker on top of a spill, evaluating other applied pressures and contact times, **propagating engineered nanomaterials from spills of dry powders (as opposed to dried spills pursued herein that were created by solvent evaporation) or from nanomaterials capped with a hydrophilic coating**, and extending the work to shear contact to simulate other worker habits.”

Specific comments –

Page 18 Line 49 – It is stated that it is likely that Cd is transferred beyond the 9th step, but it was below the LOD for ICP-MS. How were the datapoints in Figure 5 determined past the 9th secondary transfer if they were below the LOD? This should be clarified in the text.

We agree that our use of the term “detection limit” in that sentence was unclear. We have rewritten the sentence to read:

“It is likely that Cd is still transferred **in small amounts** beyond the 9th step of the simulated secondary contact events, but **the signal from these sections of countertop were**

indistinguishable from that of the pristine sections of countertop using the methods reported herein.”

Page 21 Line 49 – “This possible route for exposure should be considered in future work for guidelines and policies on workplace hygiene.” I would consider adding “and glove use” so that way it reads “policies on workplace hygiene and glove use”. I think a key concept you are fully considering is that personal protective equipment, such as gloves, if properly used and regularly changed when work is completed with nanomaterials, should limit the amount of transfer events and spread around the workplace. I think this should be emphasized here. If a worker disposes of the dirty gloves and dons a new pair, it would considerably reduce the amount of material transfer.

We have made the suggested change. We agree with the reviewer’s comment regarding the importance of frequent glove exchange and maintaining proper hygiene of PPE in the workplace. The revised text reads:

“This possible route for exposure should be considered in future work for guidelines and policies on workplace hygiene **and glove use.**”